# Regioselective Cyclic Iminium Formation of Ugi Advanced Intermediates: Rapid Access to 3,4-Dihydropyrazin-2(1*H*)-ones and Other Diverse Nitrogen-Containing Heterocycles

**DOI:** 10.3390/molecules28073062

**Published:** 2023-03-29

**Authors:** Naděžda Cankařová, Viktor Krchňák

**Affiliations:** 1Department of Organic Chemistry, Faculty of Science, Palacký University, 17 listopadu 12, 771 46 Olomouc, Czech Republic; 2Department of Chemistry and Biochemistry, 251 Nieuwland Science Center, University of Notre Dame, Notre Dame, IN 46556, USA

**Keywords:** Ugi, four-component reaction, 3,4-dihydropyrazin-2(1*H*)-ones, dipeptide, iminium, peptidomimetics, regioselectivity, diastereoselectivity, diversity-oriented synthesis

## Abstract

Herein, advanced intermediates were synthesized through Ugi four-component reactions of isocyanides, aldehydes, masked amino aldehyde, and carboxylic acids, including *N*-protected amino acids. The presence of a masked aldehyde enabled acid-mediated deprotection and subsequent cyclization via the carbonyl carbon and the amide nitrogen. Utilizing *N*-protected amino acid as a carboxylic acid component, Ugi intermediates could be cyclized from two possible directions to target 3,4-dihydropyrazin-2(1*H*)-ones. Cyclization to the amino terminus (westbound) and to the carboxyl terminus (eastbound) was demonstrated. Deliberate selection of building blocks drove the reaction regioselectively and yielded diverse heterocycles containing a 3,4-dihydropyrazin-2(1*H*)-one core, pyrazin-2(1*H*)-one, and piperazin-2-one, as well as a tricyclic framework with a 3D architecture, 2,3-dihydro-2,6-methanobenzo[*h*][1,3,6]triazonine-4,7(1*H*,5*H*)-dione, from Ugi adducts under mild reaction conditions. The latter bridged heterocycle was achieved diastereoselectively. The reported chemistry represents diversity-oriented synthesis. One common Ugi advanced intermediate was, without isolation, rapidly transformed into various nitrogen-containing heterocycles.

## 1. Introduction

Multicomponent reactions are a powerful tool for synthesizing various pharmacologically relevant compounds [1,2,3] including peptidomimetics that play an essential role in the improvement of pharmacokinetic profiles [4]. Herein, we describe a one-pot synthesis of 3,4-dihydropyrazin-2(1*H*)-ones and other diverse heterocycles from Ugi advanced intermediates. An “advanced intermediate” is a substrate with two or three diversity positions that can be used to synthesize molecular scaffolds that are structurally unrelated [5]. Dihydropyrazinone scaffolds are attractive for both organic and medicinal chemists due to their wide range of biological activities. For example, praziquantel (I) (Figure 1), which contains a piperazinone ring, is used to treat schistosomiasis [6]. More complex derivatives, including marine alkaloids, (−)-dibromophakellstatin (II), and (−)-agelastatin A (III), exhibit antineoplastic [7] and antitumor effects [8]. Hamacanthin A (IV), another representative from a class of marine alkaloids, shows antifungal properties; it acts as a growth inhibitor of *Candida albicans* [9]. A related aromatic nitrogen heterocycle, pyrazin-2(1*H*)-one, is present in aspergillic acid (V), possessing antibacterial properties [10,11]; in addition, pyrazine-2(1*H*)-one is present in dragmacidin D (VI), in which the pyrazin-2(1*H*)-one ring is between two indole rings. Dragmacidin D, a bioactive marine natural product, selectively inhibits neural nitric oxide synthase in the presence of inducible nitric oxide synthase; thus, dragmacidin D is promising in the treatment of neurodegenerative disorders such as Alzheimer’s, Parkinson’s, and Huntington’s disease [12,13,14,15].

Many protocols devoted to the traditional step-by-step synthesis of dihydropyrazinones have been published using both solid-phase synthesis [16,17,18] and solution-phase synthesis [19,20]. However, traditional step-by-step synthesis is often time-consuming and requires additional reagents for activation, expensive catalysts, etc. Multicomponent reactions (MCRs) are attractive due to their efficiency, atom and step economy, and simplicity [21,22]. MCRs are also very efficient for the synthesis of structurally complex compounds [23], which are important in the search for novel pharmacophores. Synthetic approaches utilizing MCRs that generate dihydropyrazinones and their derivatives are based on two-step procedures involving U-4CR and post-Ugi modifications. Cheng et al. prepared dihydropyrazinones using both solution- and solid-phase synthesis [24]. Different chemotypes derived from the pyrazin-2(1*H*)-one scaffold were achieved by the Ugi/deprotection/cyclization strategy [25]. Regarding saturated piperazinones, a new multicomponent approach via transition metal-catalyzed imine-directed amide N–H functionalization was described recently [26]. Concerning the fused heterocycles, regioselective formation of β- and γ-lactam-fused dihydropyrazinones was performed from the alkynoic acid-derived Ugi adducts [27]. Recently, pyrrolo[1,2-*a*]pyrazine-3,6(2*H*,4*H*)-diones were synthesized via tandem post-Ugi cyclization and gold (I)-catalyzed regioselective annulation [28]. Another approach to generate fused heterocycles involved a two-step U-4CR/allenamide cycloisomerization, leading to 6-methyl-3,4-dihydropyrazinones that were further cyclized to pyrazino[2,1-*a*]isoindoles or pyrazino[2,1-*a*]isoquinolines [29]. Fused dihydropyrazin-2(1*H*)-ones were also obtained through a U-4CR/deprotection/intramolecular Heck coupling sequence [30].

Herein, the presented synthetic strategy is based on the U-4CR of isocyanides, aldehydes, masked amino aldehyde, and carboxylic acids (including *N*-protected amino acids), which is followed by a trifluoroacetic acid (TFA)-triggered tandem reaction. As a result, the aldehyde is unmasked and then cyclized to the target 3,4-dihydropyrazin-2(1*H*)-ones without isolating the Ugi advanced intermediate. Previously, analogous model compounds were synthesized on solid supports by elongating the peptide in a stepwise manner using traditional solid-phase peptide synthesis (Figure 2) [16]. Westbound cyclization was preferable, and eastbound cyclization occurred only when westbound cyclization was not possible or disfavored.

When *N*-protected amino acid was used as the carboxylic acid component, the carbon of the unmasked carbonyl group could be subsequently attacked by two different nitrogen atoms (of different functional groups, such as amide, carbamate, and amino group) of the peptide chain (Figure 2). Thus, the following routes are possible for cyclization: so-called westbound cyclization (toward the peptide amino terminus) and eastbound cyclization (toward the carboxy terminus) [16].

In addition to 3,4-dihydropyrazin-2(1*H*)-ones **2** and **3** (Figure 1), we prepared other nitrogen-containing heterocyclic compounds from Ugi adducts. Deliberate selection of starting compounds, namely, carboxylic acids, provided access to diverse heterocycles, such as pyrazin-2(1*H*)-one **4**, piperazin-2-one **5**, and 2,3-dihydro-2,6-methanobenzo[*h*][1,3,6]triazonine-4,7(1*H*,5*H*)-dione **6**. Transformation of common intermediates to products that evince skeletal, stereochemical, or appendage diversity, so-called diversity-oriented synthesis, plays an important role in the drug discovery process [31,32,33].

## 2. Results

The target 3,4-dihydropyrazin-2(1*H*)-ones and their derivatives were prepared by one-pot three-step cyclization of Ugi adduct **1** (Figure 2). All four components of the Ugi reaction (isocyanide, aldehyde, masked amino aldehyde, and a carboxylic acid) were used in equimolar amounts and shaken in MeOH for 16 h. The resulting Ugi adduct **1** was further reacted without isolation. Ugi intermediate **1** was subsequently treated with 50% TFA in CH_2_Cl_2_, which resulted in a cascade reaction. The masked amino aldehyde was deprotected and intermediate **11** was cyclized to iminium salts **12** and **13**, which were spontaneously transformed into 3,4-dihydropyrazin-2(1*H*)-ones **2** and **3**, respectively.

Using simple commercially available building blocks, we designed and synthesized two different types of model compounds, **2** and **3** (Figure 2). The first model compounds were prepared with carboxylic acids and did not have suitable nucleophiles for westbound cyclization (benzoic acid and *p*-nitrobenzoic acid). Therefore, the product of eastbound cyclization was formed (compound **2**, Route I). Using *N*-Fmoc-protected α-amino acids (Fmoc-Gly-OH, Fmoc-Ala-OH, and Fmoc-Ser(*t*-Bu)-OH) as the carboxylic acid component, we introduced an amide into the substrate, and the westbound cyclization product was expectedly formed (compound **3**, Route II). Following our previous results [16], the reaction was regioselective, and no eastbound product was detected. In contrast, the model compound prepared using Fmoc-β-Ala-OH provided the eastbound cyclization product because the formation of a six-membered eastbound ring was preferred rather than a potential seven-membered ring (compound **2**, R’ = Fmoc–NH–(CH_2_)_2_–, Route I). When the *N*-Fmoc-protected α-amino acids were replaced by *N*-Boc derivatives (Boc-Pro-OH, Boc-Ser-OH, and Boc-Phe-OH), the TFA treatment also cleaved the Boc group and cyclization occurred between the liberated amino group and deprotected aldehyde. In those cases, cyclization of Ugi intermediate containing Boc-proline moiety resulted in the eastbound cyclization product (Route I), while cyclization of Ugi products with Boc-serine or Boc-phenylalanine moieties yielded westbound cyclization products (Route II).

Importantly, the reaction outcome was determined by the character of the amino acid protecting group (Figure 3). The Ugi reaction of isocyanide, *p*-nitrobenzaldehyde, aminoacetaldehyde dimethyl acetal, and Fmoc-Ser(*t*-Bu)-OH yielded adduct **1c**, which upon treatment with 50% TFA/CH_2_Cl_2_ resulted in *N*-Fmoc-protected 3,4-dihydropyrazin-2(1*H*)-one **3c**. In contrast, U-4CR involving Boc-Ser-OH afforded adduct **1a**. Reaction with 50% TFA/CH_2_Cl_2_ caused not only deprotection of the acetal, but also cleavage of the acid-labile Boc-protecting group (intermediate **11a**). Subsequent cyclization led to iminium salt **13a**, which underwent TFA-mediated dehydration to **14a**. Finally, spontaneous aromatization via a 1,5-hydrogen shift afforded the appropriate pyrazin-2(1*H*)-one **4a**.

In addition, we tested two other Boc-protected amino acids, Boc-Phe-OH and Boc-Pro-OH. Boc-Phe-OH was utilized for the preparation of two different heterocycles. Ugi reaction of Boc-Phe-OH with benzyl isocyanide, *p*-nitrobenzaldehyde, and aminoacetaldehyde dimethyl acetal afforded the anticipated intermediate **1b** (Figure 4), which was split into two portions. After the volatile substances were evaporated by a stream of nitrogen, the Ugi adduct was treated with two different solutions for subsequent transformations. TFA/CH_2_Cl_2_ (1:1) solution was added to the first portion. LC/MS revealed the formation of olefin intermediate **3b**. The solution was then evaporated by a stream of nitrogen, and DMSO was added for 16 h, which resulted in the formation of an oxidized product, pyrazin-1(2*H*)-one **4b**. The second part of Ugi adduct **1b** was reduced to the appropriate saturated derivative, piperazin-2-one **5a**, by treatment with TFA/triethylsilane (TES)/CH_2_Cl_2_ (5:1:4) [17]. This reagent-based approach represents westbound cyclization.

Another example of the application of Boc-protected amino acids is the reaction with Boc-Pro-OH (Figure 5). Considering the reaction outcome with Boc-Ser-OH and Boc-Phe-OH (Figure 3 and Figure 4), we expected a fused ring system to form via westbound cyclization (structure **16g**, Figure 5). However, cyclization did not follow this route and 6,7,8,8a-tetrahydropyrrolo[1,2-*a*]pyrazin-1(2*H*)-one **16g** was not formed, probably because the conformationally demanding fused ring system was avoided. Instead, cyclization occurred toward the amidic nitrogen, and 4-prolyl-3,4-dihydropyrazin-2(1*H*)-one **2g** was formed as a result of eastbound cyclization.

Unexpected but very interesting results were obtained with anthranilic acid as the carboxylic acid component. Unlike the reaction with Fmoc-β-Ala-OH, which yielded dihydropyrazinone **2f** (Table 1), anthranilic acid provided a bridged heterocycle via tandem *N*-acyliminium ion cyclization–nucleophilic addition [34,35,36]. We reported analogous reactions on unrelated substrates on several occasions for fused [16,37,38,39,40,41] and bridged heterocycles [42].

Herein, we showed the formation of a bridged scaffold using a combination of *n*-butyl isocyanide, *p*-nitrobenzaldehyde, aminoacetaldehyde dimethyl acetal, and anthranilic acid, which afforded Ugi adduct **1d**. This adduct underwent subsequent cyclization to bridged 2,3-dihydro-2,6-methanobenzo[*h*][1,3,6]triazonine-4,7(1*H*,5*H*)-dione **6a** (Figure 6). This product was a mixture of two enantiomers due to diastereoselective formation of the bridgehead chiral carbon. We expect that eastbound cyclization occurred first [43], as indicated in the scheme. However, we cannot discount an alternative reaction mechanism that first involves formation of the seven-membered ring and which is then followed by bridge ring formation.

Note that, in addition to various carboxylic acids, we tested different isocyanides and aldehydes for their compatibility with the designed synthetic routes. We chose benzyl isocyanide, *n*-butyl isocyanides, and *p*-toluenesulfonylmethyl isocyanide. However, the latter isocyanide reacted sluggishly in combination with benzoic acid, *p*-nitrobenzaldehyde, and aminoacetaldehyde dimethyl acetal. Even after three days, the reaction mixture contained a majority of a Schiff base (62%; based on LC/MS analysis). We further did not optimize the reaction. Concerning aldehydes, we tested unsubstituted benzaldehyde, *p*-nitrobenzaldehyde, *p*-cyanobenzaldehyde, and *p*-(dimethylamino)benzaldehyde. The first two mentioned aldehydes were compatible with all tested reactions, while *p*-cyanobenzaldehyde and *p*-(dimethylamino)benzaldehyde evinced some limitations. Target product **2b** (Table 1) was afforded in 90% crude HPLC purity when *p*-cyanobenzaldehyde was reacted with benzyl isocyanide, benzoic acid, and aminoacetaldehyde dimethyl acetal and underwent TFA-mediated cyclization. However, when benzoic acid was replaced with Fmoc-β-Ala-OH, the Ugi reaction with *p*-cyanobenzaldehyde failed. *p*-(Dimethylamino)benzaldehyde in reaction with benzyl isocyanide, aminoacetaldehyde dimethyl acetal, and benzoic acid yielded the expected Ugi product; nevertheless, subsequent cyclization failed. All the prepared compounds are listed in Table 1. The products were confirmed by NMR analysis, LC/MS, and HRMS. The structures of 4-prolyl-3,4-dihydropyrazin-2(1*H*)-one **2g** and bridged heterocycle **6a** were confirmed by 2D NMR experiments (COSY and HMBC); for more details, see Appendix A.

## 3. Materials and Methods

### 3.1. General Information

All used chemical reagents were purchased from commercial sources. Solvents were reagent grade and used without further purification unless stated otherwise. The LC/MS analyses were carried out using a UPLC Waters Acquity system equipped with PDA and QDa detectors. The system contained an XSelect HSS T3 (Waters) 3 mm × 50 mm C18 reverse phase column XP (2.5 μm particles). Mobile phases: 10 mM ammonium acetate in HPLC grade water (A) and gradient grade MeCN for HPLC (B). A gradient was mainly formed from 20% to 80% of B in 4.5 min and kept for 1 min, with a flow rate of 0.6 mL/min. The MS ESI operated at a 25 V cone voltage, 600 °C probe temperature, and 120 °C source temperature. Purification was carried out using semipreparative HPLC Agilent on a YMC-Actus Pro 20 mm × 100 mm C18 reversed-phase column (5 μm particles). Mobile phases: 10 mM aqueous ammonium acetate and gradient grade MeCN for HPLC at a flow rate of 15 mL/min. All ^1^H and ^13^C NMR experiments were performed at magnetic field strengths of 9.39 T (with operating frequencies of 399.78 MHz for 1H and 100.53 MHz for 13C) at ambient temperature (20 °C). In the case of compounds **3b** and **3c**, the proton measurements were performed at 80 °C. ^1^H spectra and ^13^C spectra were referenced relative to the signal of DMSO-*d*_6_ (^1^H δ = 2.50 ppm, ^13^C δ = 39.51 ppm). HRMS analyses were performed using a UPLC Dionex Ultimate 3000 equipped with an Orbitrap Elite high-resolution mass spectrometer, Thermo Exactive plus. The settings for electrospray ionization were as follows: oven temperature of 150 °C and a source voltage of 3.6 kV. The acquired data were internally calibrated with diisooctyl phthalate as a contaminant in MeOH (*m*/*z* 391.2843). UPLC separation was performed using the Phenomenex Gemini C18 column (2 mm × 50 mm, 3 μm particles). Isocratic elution was performed using the mobile phase formed of 80% MeCN and 20% buffer (10 mM ammonium acetate), and the flow rate was 0.3 mL/min.

General procedure for the synthesis of Ugi intermediate 1. Aldehyde (0.4 mmol) and carboxylic acid (0.4 mmol) were dissolved in 1 mL MeOH. Subsequently, aminoacetaldehyde dimethyl acetal (0.4 mmol) and isocyanide (0.4 mmol) were added to the solution, and the reaction mixture was stirred at room temperature for 16 h. The volatile species were then evaporated by a stream of nitrogen, and the Ugi intermediates were further used without isolation.

General procedure for the synthesis of dihydropyrazin-2(1*H*)-ones **2a–g** and **3a–c** and derivatives **4a** and **6a.** The residues containing Ugi intermediate **1** were treated with 50% TFA/CH_2_Cl_2_ (1 mL), and the solutions were shaken at room temperature for 16 h. The crude products were purified by semipreparative reversed-phase HPLC using 10 mM aq. ammonium acetate/MeCN mobile phase. MeCN was evaporated by a stream of nitrogen, and the products were freeze-dried.

Procedure for the synthesis of pyrazin-2(1*H*)-one **4b.** Ugi intermediate **1b** was treated with 50% TFA/CH_2_Cl_2_ (1 mL) at room temperature for 16 h. The volatile species were then evaporated by a stream of nitrogen, and 1 mL of DMSO was added. The reaction was shaken at room temperature for 16 h. The crude product was purified by semipreparative reversed-phase HPLC using 10 mM aq. ammonium acetate/MeCN mobile phase. MeCN was evaporated by a stream of nitrogen, and the product was freeze-dried.

Procedure for the synthesis of piperazin-2-one **5a.** Ugi intermediate **1b** was treated with 50% TFA/10% TES/CH_2_Cl_2_ (1 mL) at room temperature for 16 h. The crude product was purified by semipreparative reversed-phase HPLC using 10 mM aq. ammonium acetate/MeCN mobile phase. MeCN was evaporated by a stream of nitrogen, and the product was freeze-dried.

### 3.2. Analytical Data of Individual Compounds

4-Benzoyl-1-benzyl-3-(4-nitrophenyl)-3,4-dihydropyrazin-2(1*H*)-one **2a**



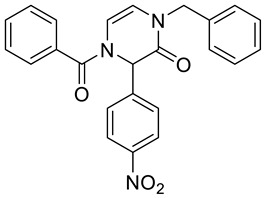



Yield: 0.114 g (69%) of amorphous solid. ESI-MS *m*/*z* = 414, [M + H]^+^. ^1^H NMR (400 MHz, DMSO-*d*_6_): δ (ppm) 8.24 (app d, *J* = 8.9 Hz, 2 H), 7.71–7.46 (m, 7 H), 7.36–7.18 (m, 5 H), 6.25 (br. s., 1 H), 6.17 (br. s., 1 H), 5.98 (br. s., 1 H), 4.82–4.69 (m, 2 H). ^13^C NMR (101 MHz, DMSO-*d*_6_): δ (ppm) 167.9, 162.2, 147.4, 143.0, 136.6, 133.2, 131.2, 128.5, 128.5, 127.6, 127.5, 127.5, 127.4, 123.8, 113.3, 110.8, 58.7, 48.4. HRMS (HESI-Orbitrap): *m*/*z* calcd. for C_24_H_20_N_3_O_4_ [M + H]^+^ 414.1448, found 414.1449.

4-(1-Benzoyl-4-benzyl-3-oxo-1,2,3,4-tetrahydropyrazin-2-yl)benzonitrile **2b**



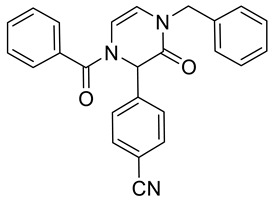



Yield: 0.108 g (69%) of amorphous solid. ESI-MS *m*/*z* = 394, [M + H]^+^. ^1^H NMR (400 MHz, DMSO-*d*_6_): δ (ppm) 7.85 (app d, *J* = 8.5 Hz, 2 H), 7.67–7.43 (m, 7 H), 7.37–7.18 (m, 5 H), 6.25 (br. s., 1 H), 6.16 (br. s., 1 H), 5.98 (br. s., 1 H), 4.84–4.67 (m, 2 H). ^13^C NMR (101 MHz, DMSO-*d*_6_): δ (ppm) 167.9, 162.3, 141.0, 136.6, 133.3, 132.7, 131.2, 131.2, 128.5, 128.5, 127.5, 127.3, 127.2, 118.4, 113.3, 116.2, 110.8, 58.8, 48.4. HRMS (HESI-Orbitrap): *m*/*z* calcd. for C_25_H_20_N_3_O_2_ [M + H]^+^ 394.1550, found 394.1546; *m*/*z* calcd. for C_25_H_18_N_3_O_2_ [M − H]^−^ 392.1394, found 392.1406.

1-Benzyl-4-(4-nitrobenzoyl)-3-(4-nitrophenyl)-3,4-dihydropyrazin-2(1*H*)-one **2c**



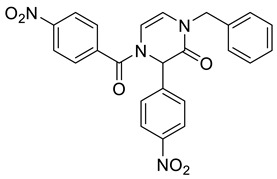



Yield: 0.112 g (61%) of amorphous solid. ESI-MS *m*/*z* = 457, [M + H]^+^. ^1^H NMR (400 MHz, DMSO-*d*_6_): δ (ppm) 8.34 (app d, *J* = 8.2 Hz, 2 H), 8.24 (app d, *J* = 8.5 Hz, 2 H), 7.87 (app d, *J* = 8.2 Hz, 2 H), 7.67 (app d, *J* = 8.5 Hz, 2 H), 7.37–7.19 (m, 5 H), 6.23 (d, *J* = 5.8 Hz, 1 H), 6.19 (s, 1 H), 6.02 (d, *J* = 5.8 Hz, 1 H), 4.76 (s, 2 H). ^13^C NMR (101 MHz, DMSO-*d*_6_): δ (ppm) 166.3, 162.1, 148.7, 147.4, 142.8, 139.4, 136.5, 130.0, 128.6, 127.7, 127.5, 127.4, 123.9, 123.7, 114.1, 109.9, 58.7, 48.5. HRMS (HESI-Orbitrap): *m*/*z* calcd. for C_24_H_17_N_4_O_6_ [M + H]^+^ 457.1143, found 457.1154.

4-Benzoyl-1-butyl-3-(4-nitrophenyl)-3,4-dihydropyrazin-2(1*H*)-one **2d**



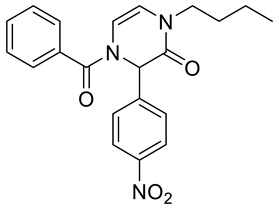



Yield: 0.060 g (40%) of amorphous solid. ESI-MS *m*/*z* = 380, [M + H]^+^. ^1^H NMR (400 MHz, DMSO-*d*_6_): δ (ppm) 8.23 (app d, *J* = 8.9 Hz, 2 H), 7.71–7.44 (m, 7 H), 6.22 (d, *J* = 4.6 Hz, 1 H), 6.11 (br. s., 1 H), 5.98 (d, *J* = 4.6 Hz, 1 H), 3.60–3.45 (m, 2 H), 1.56–1.43 (m, 2 H), 1.27–1.14 (m, 2 H), 0.84 (t, *J* = 7.3 Hz, 3 H). ^13^C NMR (101 MHz, DMSO-*d*_6_): δ (ppm) 167.8, 161.8, 147.3, 143.1, 133.3, 131.2, 128.5, 127.5, 123.8, 113.6, 110.3, 93.1, 58.6, 45.1, 29.7, 19.1, 13.4. HRMS (HESI-Orbitrap): *m*/*z* calcd. for C_21_H_22_N_3_O_4_ [M + H]^+^ 380.1605, found 380.1602; *m*/*z* calcd. for C_21_H_20_N_3_O_4_ [M − H]^−^ 378.1448, found 378.1457.

1-Butyl-4-(4-nitrobenzoyl)-3-(4-nitrophenyl)-3,4-dihydropyrazin-2(1*H*)-one **2e**



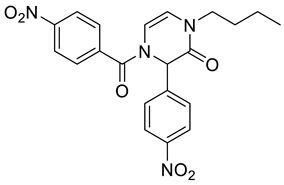



Yield: 0.049 g (29%) of amorphous solid. ESI-MS *m*/*z* = 423, [M − H]^−^. ^1^H NMR (400 MHz, DMSO-*d*_6_): δ (ppm) 8.36 (app d, *J* = 8.5 Hz, 2 H), 8.25 (app d, *J* = 8.5 Hz, 2 H), 7.87 (app d, *J* = 8.5 Hz, 2 H), 7.66 (app d, *J* = 8.5 Hz, 2 H), 6.17 (d, *J* = 5.8 Hz, 1 H), 6.09 (s, 1 H), 6.02 (d, *J* = 5.8 Hz, 1 H), 3.52 (t, *J* = 7.0 Hz, 2 H), 1.54–1.44 (m, 2 H), 1.22 (qd, *J* = 7.1, 14.6 Hz, 2 H), 0.85 (t, *J* = 7.3 Hz, 3 H). ^13^C NMR (101 MHz, DMSO-*d*_6_): δ (ppm) 166.2, 161.8, 148.7, 147.4, 142.9, 139.4, 129.9, 127.6, 123.9, 123.7, 114.4, 109.4, 58.6, 45.2, 29.7, 19.1, 13.5. HRMS (HESI-Orbitrap): *m*/*z* calcd. for C_21_H_21_N_4_O_6_ [M + H]^+^ 425.1456, found 425.1457; *m*/*z* calcd. for C_21_H_19_N_4_O_6_ [M − H]^−^ 423.1299, found 423.1313.

(9*H*-Fluoren-9-yl)methyl (3-(4-butyl-2-(4-nitrophenyl)-3-oxo-3,4-dihydropyrazin-1(2*H*)-yl)-3-oxopropyl)carbamate **2f**



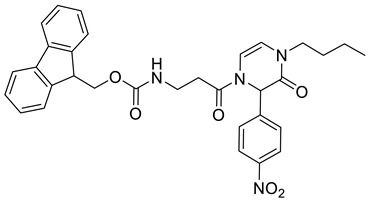



Yield: 0.089 g (40%) of amorphous solid. ESI-MS *m*/*z* = 569, [M + H]^+^. ^1^H NMR (400 MHz, DMSO-*d*_6_): δ (ppm) 8.25–8.14 (m, 2 H), 7.87 (d, *J* = 7.6 Hz, 2 H), 7.72–7.62 (m, 2 H), 7.54 (d, *J* = 8.9 Hz, 2 H), 7.44–7.24 (m, 5 H), 6.61 (d, *J* = 5.8 Hz, 1 H), 6.09–5.98 (m, 2 H), 4.32 (d, *J* = 6.7 Hz, 2 H), 4.27–4.14 (m, 1 H), 3.55–3.41 (m, 2 H), 3.31 (td, *J* = 6.9, 13.6 Hz, 2 H), 2.89–2.78 (m, 1 H), 2.77–2.65 (m, 1 H), 1.53–1.37 (m, 2 H), 1.18 (qd, *J* = 7.3, 14.6 Hz, 2 H), 0.89–0.75 (m, 3 H). ^13^C NMR (101 MHz, DMSO-*d*_6_): δ (ppm) 169.6, 162.0, 156.1, 147.2, 143.9, 143.4, 140.7, 127.5, 127.5, 127.0, 125.1, 123.7, 120.0, 114.2, 108.5, 65.3, 57.3, 46.7, 45.1, 36.2, 32.8, 29.7, 19.1, 13.4. HRMS (HESI-Orbitrap): *m*/*z* calcd. for C_32_H_33_N_4_O_6_ [M + H]^+^ 569.2395, found 569.2396; *m*/*z* calcd. for C_32_H_31_N_4_O_6_ [M − H]^−^ 567.2238, found 567.2248.

4-(L-Prolyl)-1-benzyl-3-(4-nitrophenyl)-3,4-dihydropyrazin-2(1*H*)-one **2g**



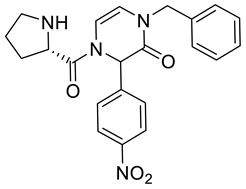



Yield: 0.098 g (60%) of amorphous solid of a mixture of two diastereomers. ESI-MS *m*/*z* = 407, [M + H]^+^. ^1^H NMR (400 MHz, DMSO-*d*_6_): δ (ppm) 8.30–8.15 (m, 4 H), 7.62–7.47 (m, 4 H), 7.35–7.25 (m, 6 H), 7.25–7.18 (m, 4 H), 6.89–6.82 (m, 1 H), 6.76–6.72 (m, 1 H), 6.12–6.01 (m, 4 H), 4.86–4.75 (m, 2 H), 4.69–4.59 (m, 2 H), 4.19–4.11 (m, 1 H), 4.09–4.02 (m, 1 H), 4.00–3.70 (m, 2 H), 2.95–2.80 (m, 2 H), 2.79–2.66 (m, 2 H), 2.16–2.03 (m, 1 H), 1.98–1.86 (m, 1 H), 1.83–1.73 (m, 1 H), 1.73–1.56 (m, 5 H). ^13^C NMR (101 MHz, DMSO-*d*_6_): δ (ppm) 172.6, 162.3, 162.2, 147.3, 143.5, 143.1, 136.6, 136.6, 128.5, 127.5, 127.4, 127.3, 123.9, 123.8, 113.9, 113.5, 109.2, 108.8, 57.6, 57.5, 48.5, 48.5, 47.0, 46.8, 29.1, 28.9, 26.0, 26.0. HRMS (HESI-Orbitrap): *m*/*z* calcd. for C_22_H_23_N_4_O_4_ [M + H]^+^ 407.1714, found 407.1711; *m*/*z* calcd. for C_22_H_21_N_4_O_4_ [M − H]^−^ 405.1557, found 405.1569.

(9*H*-Fluoren-9-yl)methyl 4-(2-(butylamino)-1-(4-nitrophenyl)-2-oxoethyl)-3-oxo-3,4-dihydropyrazine-1(2*H*)-carboxylate **3a**



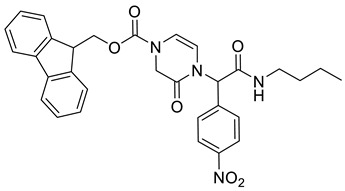



Yield: 0.075 g (44%) of amorphous solid. ESI-MS *m*/*z* = 555, [M + H]^+^. ^1^H NMR (400 MHz, DMSO-*d*_6_): δ (ppm) 8.53 (t, *J* = 5.0 Hz, 1 H), 8.25 (app d, *J* = 8.5 Hz, 2 H), 7.89 (d, *J* = 5.5 Hz, 2 H), 7.63 (d, *J* = 6.1 Hz, 2 H), 7.53 (app d, *J* = 8.2 Hz, 2 H), 7.46–7.38 (m, 2 H), 7.33 (d, *J* = 6.1 Hz, 2 H), 6.33–6.22 (m, 1 H), 6.11 (br. s., 1 H), 5.69 (br. s., 1 H), 4.47 (br. s., 2 H), 4.32 (d, *J* = 4.9 Hz, 1 H), 4.28–4.18 (m, 2 H), 3.22–3.07 (m, 2 H), 1.48–1.36 (m, 2 H), 1.28 (qd, *J* = 7.3, 14.8 Hz, 2 H), 0.87 (t, *J* = 7.3 Hz, 3 H). ^13^C NMR (101 MHz, DMSO-*d*_6_): δ (ppm) 166.6, 147.2, 143.5, 143.1, 140.7, 129.7, 127.7, 127.2, 125.0, 123.7, 120.2, 110.4, 108.5, 67.5, 57.3, 46.4, 43.3, 38.5, 30.8, 19.5, 13.5. HRMS (HESI-Orbitrap): *m*/*z* calcd. for C_31_H_31_N_4_O_6_ [M + H]^+^ 555.2238, found 555.2244; *m*/*z* calcd. for C_31_H_29_N_4_O_6_ [M − H]^−^ 553.2082, found 553.2097.

(9*H*-Fluoren-9-yl)methyl (2*S*)-4-(2-(butylamino)-1-(4-nitrophenyl)-2-oxoethyl)-2-methyl-3-oxo-3,4-dihydropyrazine-1(2*H*)-carboxylate **3b**



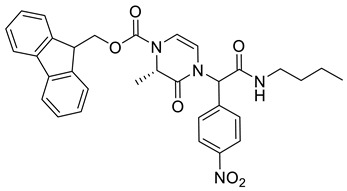



Yield: 0.102 g (45%) of amorphous solid of a mixture of two diastereomers. ESI-MS *m*/*z* = 569, [M + H]^+^. ^1^H NMR (400 MHz, DMSO-*d*_6_): δ (ppm) 8.29–8.19 (m, 6 H), 7.90–7.83 (m, 4 H), 7.66–7.58 (m, 4 H), 7.51 (d, *J* = 8.2 Hz, 4 H), 7.45–7.38 (m, 4 H), 7.37–7.28 (m, 4 H), 6.25 (s, 1 H), 6.22 (s, 1 H), 6.07–5.98 (m, 2 H), 5.87 (d, *J* = 6.1 Hz, 1 H), 5.68 (d, *J* = 6.1 Hz, 1 H), 4.65–4.55 (m, 4 H), 4.55–4.44 (m, 2 H), 4.34 (q, *J* = 6.1 Hz, 2 H), 3.21–3.11 (m, 4 H), 1.49–1.39 (m, 4 H), 1.35–1.23 (m, 4 H), 1.11–1.00 (m, 6 H), 0.92–0.83 (m, 6 H). ^13^C NMR (101 MHz, DMSO-*d*_6_): δ (ppm) 166.7, 166.4, 165.3, 165.1, 151.8, 151.6, 147.3, 147.2, 143.6, 143.2, 143.0, 140.8, 129.7, 129.4, 127.7, 127.1, 125.0, 124.7, 123.9, 123.8, 120.1, 111.1, 110.5, 67.3, 67.1, 57.8, 57.1, 53.0, 52.6, 46.6, 38.6, 38.5, 30.8, 30.8, 19.5, 19.4, 15.1, 15.0, 13.5. HRMS (HESI-Orbitrap): *m*/*z* calcd. for C_32_H_33_N_4_O_6_ [M + H]^+^ 569.2395, found 569.2400; *m*/*z* calcd. for C_32_H_31_N_4_O_6_ [M − H]^−^ 567.2238, found 567.2249.

(9*H*-Fluoren-9-yl)methyl (2*S*)-4-(2-(benzylamino)-1-(4-nitrophenyl)-2-oxoethyl)-2-(hydroxymethyl)-3-oxo-3,4-dihydropyrazine-1(2*H*)-carboxylate **3c**



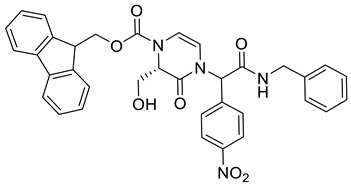



Yield: 0.078 g (31%) of amorphous solid of a mixture of two diastereomers. ESI-MS *m*/*z* = 619, [M + H]^+^. ^1^H NMR (400 MHz, DMSO-*d*_6_): δ (ppm) 8.82 (br. s., 1 H), 8.73 (br. s., 1 H), 8.25–8.15 (m, 4 H), 7.91–7.84 (m, 4 H), 7.69–7.61 (m, 4 H), 7.58 (dd, *J* = 5.2, 8.5 Hz, 4 H), 7.45–7.38 (m, 4 H), 7.37–7.22 (m, 14 H), 6.42–6.34 (m, 2 H), 6.26–6.20 (m, 2 H), 5.85 (d, *J* = 6.1 Hz, 1 H), 5.58 (d, *J* = 6.1 Hz, 1 H), 4.97–4.88 (m, 2 H), 4.70–4.56 (m, 2 H), 4.55–4.47 (m, 4 H), 4.43–4.38 (m, 4 H), 4.37–4.30 (m, 2 H), 3.73–3.46 (m, 4 H). HRMS (HESI-Orbitrap): *m*/*z* calcd. for C_35_H_31_N_4_O_7_ [M + H]^+^ 619.2187, found 619.2185; *m*/*z* calcd. for C_35_H_29_N_4_O_7_ [M − H]^−^ 617.2031, found 617.2042.

*N*-Butyl-2-(3-methyl-2-oxopyrazin-1(2*H*)-yl)-2-(4-nitrophenyl)acetamide **4a**



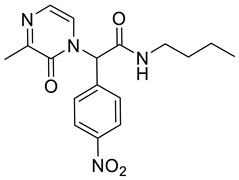



Yield: 0.015 g (11%) of amorphous solid. ESI-MS *m*/*z* = 345, [M + H]^+^. ^1^H NMR (400 MHz, DMSO-*d*_6_): δ (ppm) 8.71 (t, *J* = 5.5 Hz, 1 H), 8.29 (app d, *J* = 8.9 Hz, 2 H), 7.59 (app d, *J* = 8.5 Hz, 2 H), 7.12–7.09 (m, 1 H), 7.08–7.05 (m, 1 H), 6.65 (s, 1 H), 3.23–3.07 (m, 2 H), 2.33 (s, 3 H), 1.46–1.37 (m, 2 H), 1.26 (qd, *J* = 7.3, 14.8 Hz, 2 H), 0.86 (t, *J* = 7.3 Hz, 3 H). ^13^C NMR (101 MHz, DMSO-*d*_6_): δ (ppm) 165.7, 156.1, 155.0, 147.6, 142.0, 130.4, 126.0, 124.1, 121.5, 59.7, 38.7, 30.7, 20.7, 19.4, 13.5. HRMS (HESI-Orbitrap): *m*/*z* calcd. for C_17_H_21_N_4_O_4_ [M + H]^+^ 345.1557, found 345.1554; *m*/*z* calcd. for C_17_H_19_N_4_O_4_ [M − H]^−^ 343.1401, found 343.1411.

*N*-Benzyl-2-(3-benzyl-2-oxopyrazin-1(2*H*)-yl)-2-(4-nitrophenyl)acetamide **4b**



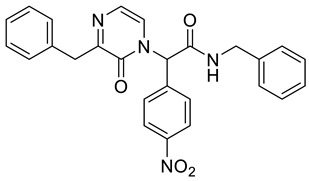



Yield: 0.025 g (27%) of amorphous solid. ESI-MS *m*/*z* = 455, [M + H]^+^. ^1^H NMR (400 MHz, DMSO-*d*_6_): δ (ppm) 9.22 (t, *J* = 5.6 Hz, 1 H), 8.27 (app d, *J* = 8.5 Hz, 2 H), 7.61 (app d, *J* = 8.9 Hz, 2 H), 7.34–7.22 (m, 10 H), 7.17 (d, *J* = 4.6 Hz, 1 H), 7.11 (d, *J* = 4.6 Hz, 1 H), 6.71 (s, 1 H), 4.37 (d, *J* = 5.8 Hz, 2 H), 4.05 (d, *J* = 4.9 Hz, 2 H). ^13^C NMR (101 MHz, DMSO-*d*_6_): δ (ppm) 166.0, 157.5, 154.7, 147.7, 141.4, 138.3, 137.6, 130.6, 129.2, 128.3, 128.2, 127.4, 127.0, 126.4, 126.2, 124.1, 121.7, 60.3, 42.7. HRMS (HESI-Orbitrap): *m*/*z* calcd. for C_26_H_23_N_4_O_4_ [M + H]^+^ 455.1714, found 455.1716.

*N*-Benzyl-2-((*S*)-3-benzyl-2-oxopiperazin-1-yl)-2-(4-nitrophenyl)acetamide **5a**



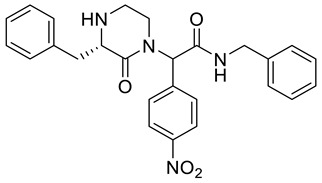



Yield: 0.053 g (58%) of amorphous solid of a mixture of two diastereomers. ESI-MS *m*/*z* = 459, [M + H]^+^. ^1^H NMR (400 MHz, DMSO-*d*_6_): δ (ppm) 8.87 (t, *J* = 6.1 Hz, 1 H), 8.84 (t, *J* = 5.8 Hz, 1 H), 8.23 (app d, *J* = 8.5 Hz, 2 H), 8.19 (app d, *J* = 8.9 Hz, 2 H), 7.51 (d, *J* = 8.9 Hz, 2 H), 7.37–7.15 (m, 22 H), 6.36 (s, 1 H), 6.34 (s, 1 H), 4.43–4.30 (m, 4 H), 3.68 (dd, *J* = 3.7, 7.9 Hz, 1 H), 3.62 (dd, *J* = 3.7, 9.2 Hz, 1 H), 3.50–3.42 (m, 1 H), 3.37–3.30 (m, 2 H), 3.22 (dd, *J* = 3.5, 13.9 Hz, 1 H), 3.13 (dd, *J* = 3.5, 13.6 Hz, 1 H), 2.98–2.60 (m, 9 H). ^13^C NMR (101 MHz, DMSO-*d*_6_): δ (ppm) 170.0, 170.0, 168.1, 167.9, 147.0, 147.0, 143.6, 143.2, 139.1, 139.0, 138.9, 138.8, 130.2, 129.9, 129.6, 128.4, 129.3, 128.3, 128.1, 128.0, 127.3, 127.3, 126.9, 126.0, 126.0, 123.5, 123.4, 60.3, 60.2, 59.3, 58.6, 45.9, 45.8, 42.3, 42.3, 41.2, 41.1, 38.0, 37.9. HRMS (HESI-Orbitrap): *m*/*z* calcd. for C_26_H_27_N_4_O_4_ [M + H]^+^ 459.2027, found 459.2024; *m*/*z* calcd. for C_26_H_25_N_4_O_4_ [M − H]^−^ 457.1870, found 457.1883.

(2*R*,5*R*,6*S*)-3-Butyl-5-(4-nitrophenyl)-2,3-dihydro-2,6-methanobenzo[*h*][1,3,6]triazonine-4,7(1*H*,5*H*)-dione and (2*S*,5*S*,6*R*)-3-butyl-5-(4-nitrophenyl)-2,3-dihydro-2,6-methanobenzo[*h*][1,3,6]triazonine-4,7(1*H*,5*H*)-dione **6a**



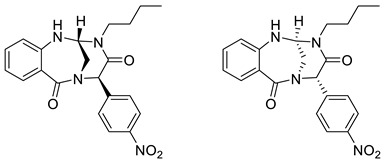



Yield: 0.016 g (10%) of amorphous solid. ESI-MS *m*/*z* = 395, [M + H]^+^. ^1^H NMR (400 MHz, DMSO-*d*_6_): δ (ppm) 8.30 (app d, *J* = 8.5 Hz, 2 H), 7.75 (app d, *J* = 8.5 Hz, 2 H), 7.55 (d, *J* = 6.7 Hz, 1 H), 7.40 (d, *J* = 6.4 Hz, 1 H), 7.36–7.28 (m, 1 H), 6.85–6.74 (m, 2 H), 6.03 (s, 1 H), 5.07 (dd, *J* = 3.1, 6.1 Hz, 1 H), 3.72 (d, *J* = 15.3 Hz, 1 H), 3.59 (td, *J* = 7.7, 13.7 Hz, 1 H), 3.44 (dd, *J* = 3.4, 15.3 Hz, 1 H), 3.29–3.19 (m, 1 H), 1.59 (br. s., 2 H), 1.31 (qd, *J* = 7.1, 13.9 Hz, 2 H), 0.93 (t, *J* = 7.3 Hz, 3 H). ^13^C NMR (101 MHz, DMSO-*d*_6_): δ (ppm) 172.7, 164.0, 147.2, 144.1, 143.0, 132.8, 131.3, 128.9, 123.8, 119.2, 118.1, 118.0, 67.9, 62.2, 44.0, 41.4, 29.0, 19.8, 13.8. HRMS (HESI-Orbitrap): *m*/*z* calcd. for C_21_H_23_N_4_O_4_ [M + H]^+^ 395.1714, found 395.1712; *m*/*z* calcd. for C_21_H_21_N_4_O_4_ [M − H]^−^ 393.1557, found 393.1570.

## 4. Conclusions

To conclude, we synthesized Ugi advanced intermediates and achieved straightforward transformations into a diversity of nitrogen-containing heterocycles. The direction of cyclization was dependent on the character (structure, carbon chain length, presence of other nucleophile and protecting groups) of the starting carboxylic acid. Different reaction outcomes were obtained when carboxylic acids without a nucleophilic functional group (benzoic acid and *p*-nitrobenzoic acid) or amino acids with different lengths of carbon chains and the character of the protecting groups (Fmoc-Gly-OH, Fmoc-Ala-OH, Fmoc-β-Ala-OH, Fmoc-Ser(*t*-Bu)-OH, Boc-Ser-OH, Boc-Phe-OH, Boc-Pro-OH, and anthranilic acid) were incorporated into the Ugi intermediate. First, the length of the carbon chain was crucial. Six-membered rings were favorable (with Fmoc-Gly-OH, Fmoc-Ala-OH, Fmoc-Ser(*t*-Bu)-OH) and westbound cyclization of Ugi intermediates occurred, while reaction with Fmoc-β-Ala-OH resulted in eastbound cyclization. Second, the character of the amino acid protecting group determined the formation of the target product. While Fmoc-protected Ser(*t*-Bu)-OH was included in the Ugi adduct, TFA-mediated cyclization resulted in the formation of 3,4-dihydropyrazin-2(1*H*)-ones. In contrast, Ugi reaction with Boc-protected Ser-OH resulted in cyclization and spontaneous dehydration followed by aromatization to pyrazin-2(1*H*)-one. We also reduced the dihydropyrazinone cycle to piperazinone through TFA/TES/CH_2_Cl_2_ treatment. The Ugi intermediate containing an anthranilic acid moiety resulted in tandem diastereoselective cyclization to a bridged heterocycle with a 3D architecture.

## Data Availability

Not applicable.

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
