# Peer review of "Regioselective Cyclic Iminium Formation of Ugi Advanced Intermediates: Rapid Access to 3,4-Dihydropyrazin-2(1H)-ones and Other Diverse Nitrogen-Containing Heterocycles"

_molecules, 2023, doi:10.3390/molecules28073062_

Round 1

Reviewer 1 Report

The manuscript by Cankarova and Krchnak presents a new regioselective synthesis of advanced intermediates through the four-component Ugi reaction. The work provides access to significantly complex chemical structures and contributes to diversifying the scope of the Ugi reaction. The presented work is essential as the proposed reaction simplifies the approach to synthesizing complex natural products and allows for the straightforward, time-efficient development of compound libraries. The overall quality of the manuscript is high. It is well-structured, well-written, and supplemented by sufficient high-quality graphics. The experimental section produces adequate technical description and experiment design. The compound characterization is sufficient. The Reviewer did not identify any significant flaws. The recommendation is "accept in present form".

Reviewer 2 Report

The manuscript submitted by CankaÅ™ová and Krchňák presents a comprehensive study of a multicomponent reaction for obtaining nitrogenated heterocycles.

Overall, the manuscript is interesting, one-step multicomponent reactions such as those presented by the authors are of current relevance as they respond to organic chemistry needs such as the synthesis of new drugs. In that sense, the topic fits perfectly with the journal Molecules. 

To this reviewer, it seems a very interesting topic but he has found it tremendously difficult to follow up the manuscript.

First of all, the quality of the English must be highly revised. There are many poorly worded expressions and some words that are not well contextualized. Still, the biggest problem is the organization of the synthetic methodology used. 

In the case of Scheme 1, the authors number compounds 5, 8, 8g, 10, 14-16. This is not correct. I understand that the authors want to illustrate their objective in a scheme but never number only some compounds without having numbered 1-4, 6-7,...etc. Also in this scheme, I believe that it should appear, even if only in an exemplary way, that they are R1-R4.

It is necessary that the authors redo this schema correctly. 

Scheme 1 appears in the manuscript without being previously cited. 

In line 109 I believe that they cite Scheme 4 without having named Schemes 2 and 3. These are very serious flaws in a scientific manuscript. 

Regarding the schemes themselves, it has been very difficult for this reviewer to know what are the radicals indicated as R1-R4. It is also not clear what the brackets in the molecules mean, are they molecules that have not been isolated or purified? The meaning of Route I-III should be further explained within the text itself. 

The concrete structures of the molecules should be able to be deduced from the figure itself and not having to go to the experimental part to check the structure referred to by the authors. 

In scheme 3. What does it mean that these three molecules are between []?

At what point is the TFA added? This scheme seems totally incorrect. The same situation is found in Scheme 4, 5 and 6.

Finally, in a table, there is a detail of some of the structures in the manuscript. This reviewer does not understand that a series of compounds appear above this table. Compound 8 in general, 8g in particular, and 10, 14-16 in general. The figure above the table does not contribute anything and only confuses. What does the column "entry" mean? In necessary?

Also the experimental part has serious problems. In the description of the multiplicity in 1H (s, d, t...) none of these terms are italicized as they should be. Otherwise, for example, "s" without italics is second and not singlet. 

The authors have put all the 13C signals in a list that does not contribute anything. It would be better to indicate the most representative signals with reference to the part of the molecule they want to indicate. 

Finally note that the introduction begins by stating: "Among them, peptidomimetics play an essential role 31 as they have improved pharmacokinetic profiles compared to peptides [4]. Herein, we 32 describe a one-pot synthesis of 3,4-dihydropyrazin-2(1H)-ones and other diverse hetero-33 cycles from Ugi advanced intermediates." and never again in the manuscript is there any allusion to peptidomimetics or whether the compounds have been designed in that sense on the basis of radicals or, alternatively, for their availability or reactivity.

Overall, this reviewer believes that the manuscript has a good thematic orientation but that the organization of the results as well as the formal aspects need major revision for publication in the journal molecules. 

Reviewer 3 Report

In this article, the authors describe the synthesis of different nitrogen heterocycles using the multicomponent Ugi reaction. The reaction is very versatile and allows them to control the new compound depending on the components used in the reaction. In this article, 14 products have been synthesized, with yields ranging from 10 to 69%, but considering the number of steps involved, this is a good result.

The article is written in a concise and organized manner and is easy to read and follow. The schemes, figures and table are thoroughly detailed. The introduction and reference section are adequate and up to date (however, reference 10 should be checked for more recent similar one). Finally the experimental section is suitable.

Although I have several comments:

In the results section: Scheme 2 in route I, in b) is missing R2?

In Table 1 it can be noticed that the yields of compounds containing an n-Bu group are lower than those containing a benzyl group, but there is no comment about this in the text. It is possible that due to the extraction process the products are too volatile and go away with the nitrogen stream?

And then in the experimental section, compounds 8g, 10b, 10c, and 15a are said to be obtained as a mixture of two diastereoisomers although I understood that they are in equimolecular ratio (and NMR proved it), it could be mentioned as in 1/1 ratio in the description.

Author Response

Please see tha attachment.

Round 2

Reviewer 2 Report

The version of the manuscript received has significantly improved the quality of the research presented.

The manuscript can be accepted in the present form.